# Phase Transitions and Cyclic Phenomena in Bandits with Switching Constraints

**David Simchi-Levi**
Institute for Data, Systems and Society
Massachusetts Institute of Technology
Cambridge, MA 02139
dslevi@mit.edu

**Yunzong Xu**
Institute for Data, Systems and Society
Massachusetts Institute of Technology
Cambridge, MA 02139
yxu@mit.edu

## Abstract

We consider the classical stochastic multi-armed bandit problem with a constraint on the total cost incurred by switching between actions. Under the unit switching cost structure, where the constraint limits the total number of switches, we prove matching upper and lower bounds on regret and provide near-optimal algorithms for this problem. Surprisingly, we discover phase transitions and cyclic phenomena of the optimal regret. That is, we show that associated with the multi-armed bandit problem, there are equal-length phases defined by the number of arms and switching costs, where the regret upper and lower bounds in each phase remain the same and drop significantly between phases. The results enable us to fully characterize the trade-off between regret and incurred switching cost in the stochastic multi-armed bandit problem, contributing new insights to this fundamental problem. Under the general switching cost structure, our analysis reveals a surprising connection between the bandit problem and the shortest Hamiltonian path problem.

## 1   Introduction

The multi-armed bandit (MAB) problem is one of the most fundamental problems in online learning, with diverse applications ranging from pricing and online advertising to clinical trails. In a traditional MAB problem, the learner (i.e., decision-maker) is allowed to switch freely between actions, and an effective learning policy may incur frequent switching — indeed, the learner's task is to balance the exploration-exploitation trade-off, and both exploration (i.e., acquiring new information) and exploitation (i.e., optimizing decisions based on up-to-date information) require switching. However, in many real-world scenarios, it is costly to switch between different alternatives, and a learning policy with limited switching behavior is preferred. The learner thus has to consider the cost of switching in her learning task.

**The conventional model: switching cost as a penalty.** There is rich literature studying stochastic MAB with switching costs. Most of the papers model the switching cost as a penalty in the learner's objective, i.e., they measure a policy's regret and incurred switching cost using the same metric and the objective is to minimize the sum of these two terms (e.g., [1, 2, 7, 8]; there are other variations with discounted rewards [5, 4, 6], see [12] for a survey). Though this conventional "switching penalty" model has attracted significant research interest in the past, it has two limitations. First, under this model, the learner's total switching cost is a complete output determined by the learning algorithm. However, in many real-world applications, there are strict limits on the learner's switching behavior, which should be modeled as a *hard constraint*, and hence the learner's total budget of switching cost should be an input that helps determine the algorithm. In particular, while the algorithm in [8] developed for the "switching penalty" model can achieve $\tilde{O}(\sqrt{T})$ (distribution-free) regret with $O(\log \log T)$ switches, if the learner wants a policy that always incurs finite switching cost

independent of $T$, then prior literature does not provide an answer. Second, the "switching penalty" model has fundamental weakness in studying the trade-off between regret and incurred switching cost in stochastic MAB — since the $O(\log \log T)$ bound on the incurred switching cost of a policy is negligible compared with the $\tilde{O}(\sqrt{T})$ bound on its optimal regret, when adding the two terms up, the term associated with incurred switching cost is always dominated by the regret, thus no trade-off can be identified. As a result, to the best of our knowledge, prior literature has not characterized the fundamental trade-off between regret and incurred switching cost in stochastic MAB.

**The BwSC model: switching as a limited resource.** In this paper, we introduce the *Bandits with Switching Constraints* (BwSC) problem. The BwSC model addresses the issues associated with the "switching penalty" model in several ways. First, it introduces a hard constraint on the total switching cost, making the *switching budget* an input to learning policies, enabling us to design good policies that guarantee limited switching cost. While $O(\log \log T)$ switches has proven to be sufficient for a learning policy to achieve near-optimal regret in MAB, in BwSC, we are mostly interested in the setting of finite or $o(\log \log T)$ switching budget, which is highly relevant in practice. Second, by focusing on rewards in the objective function and incurred switching cost in the switching constraint, the BwSC framework enables the characterization of the fundamental trade-off between regret and maximum incurred switching cost in MAB. Third, while most prior research assumes specific structures on switching costs (e.g., unit or homogeneous costs), in reality, switching between different pairs of actions may incur heterogeneous costs that do not follow any parametric form. The BwSC model allows general switching costs, which makes it a powerful modeling framework.

**Motivating examples.** The BwSC framework has numerous applications, including dynamic pricing, online assortment optimization, online advertising, clinical trails and vehicle routing. A representative example is the *dynamic pricing* problem. Dynamic pricing with demand learning has proven its effectiveness in online retailing. However, it is well known that in practice, sellers often face business constraints that prevent them from conducting extensive price experimentation and making frequent price changes. For example, according to [10], Groupon limits the number of price changes, either because of implementation constraints, or for fear of confusing customers and receiving negative customer feedback. In such scenarios, the seller's sequential decision-making problem can be modeled as a BwSC problem, where changing from each price to another price incurs some cost, and there is a limit on the total cost incurred by price changes.

**Main contributions.** In this paper, we introduce the BwSC model, a general framework with strong modeling power. The model overcomes the limitations of the prior "switching penalty" model and has both practical and theoretical values.

We first study the unit-switching-cost BwSC problem in Section 4. We develop an upper bound on regret by proposing a simple and intuitive policy with carefully-designed switching rules, and prove an information-theoretic lower bound that matches the above upper bound, indicating that our policy is rate-optimal up to logarithmic factors. Methodologically, the proof of the lower bound involves a novel "tracking the cover time" argument that has not appeared in prior literature and may be of independent interest. With the analysis described above we obtain some surprising and insightful results, namely, phase transitions and cyclic phenomena of the optimal regret. That is, we show that associated with the BwSC problem, there are equal-length phases, defined by the number of arms and switching costs, where the regret upper and lower bounds in each phase remain the same and drop significantly between phases, see the precise definitions in Section 4.3.

We then study the general-switching-cost BwSC problem in Section 5. We propose an efficient policy and prove regret upper and lower bounds in the general setting. The results reveal a surprising connection between the BwSC problem and the shortest Hamiltonian path problem.

For the full version of this paper (containing additional results and missing proofs), see [14].

## 2   Notations, Model and Definitions

**Notations.** For all $n_1, n_2 \in \mathbb{N}$ such that $n_1 \leq n_2$, we use $[n_1]$ to denote the set $\{1, \ldots, n_1\}$, and use $[n_1 : n_2]$ (resp. $(n_1 : n_2)$) to denote the set $\{n_1, n_1 + 1, \ldots, n_2\}$ (resp. $\{n_1 + 1, \ldots, n_2\}$). For all $x \geq 0$, we use $\lfloor x \rfloor$ to denote the largest integer less than or equal to $x$. For ease of presentation, we define $\lfloor x \rfloor = 0$ for all $x < 0$. Throughout the paper, we use big $O, \Omega, \Theta$ notations to hide constant factors, and use $\tilde{O}, \tilde{\Omega}, \tilde{\Theta}$ notations to hide constant factors and logarithmic factors.

**Problem formulation.** Consider a $k$-armed bandit problem where a learner chooses actions from a fixed set $[k] = \{1, \ldots, k\}$. There is a total of $T$ rounds. In each round $t \in [T]$, the learner first chooses an action $i_t \in [k]$, then observes a reward $r_t(i_t) \in \mathbb{R}$. For each action $i \in [k]$, the reward of action $i$ is i.i.d. drawn from an (unknown) distribution $\mathcal{D}_i$ with (unknown) expected value $\mu_i$. We assume that the distributions $\mathcal{D}_i$ are standardized sub-Gaussian. Without loss of generality, we assume $\sup_{i,j \in [k]} |\mu_i - \mu_j| \in [0, 1]$.

In our problem, the learner incurs a switching cost $c_{i,j} = c_{j,i} \geq 0$ each time she switches between action $i$ and action $j$ $(i, j \in [k])$. In particular, $c_{i,i} = 0$ for $i \in [k]$. There is a pre-specified *switching budget* $S \geq 0$ representing the maximum amount of switching costs that the learner can incur in total. Once the total switching cost exceeds the switching budget $S$, the learner cannot switch her actions any more. The learner's goal is to maximize the expected total reward over $T$ rounds.

**Admissible policies.** Let $\pi$ denote the learner's (non-anticipating) learning policy, and $\pi_t \in [k]$ denote the action chosen by policy $\pi$ at round $t \in [T]$. More formally, $\pi_t$ establishes a probability kernel acting from the space of historical actions and observations to the space of actions at round $t$. Let $\mathbb{P}_{\mathcal{D}}^{\pi}$ and $\mathbb{E}_{\mathcal{D}}^{\pi}$ be the probability measure and expectation induced by policy $\pi$ and latent distributions $\mathcal{D} = (\mathcal{D}_1, \ldots, \mathcal{D}_k)$. According to the problem formulation, we only need to restrict our attention to the *S-switching-budget* policies, which take $S$, $k$ and $T$ as input and are defined below.

**Definition 1** *A policy $\pi$ is said to be an S-switching-budget policy if for all $\mathcal{D}$,*

$$\mathbb{P}_{\mathcal{D}}^{\pi}\left[\sum_{t=1}^{T-1} c_{\pi_t, \pi_{t+1}} \leq S\right] = 1.$$

Let $\Pi_S$ denote the set of all $S$-switching-budget policies, which is also the admissible policy class of the BwSC problem.

**Regret.** The performance of a learning policy is measured against a clairvoyant policy that maximizes the expected total reward given foreknowledge of the environment (i.e., latent distributions) $\mathcal{D}$. Let $\mu^* = \max_{i \in [k]} \mu_i$. We define the *regret* of policy $\pi$ as the worst-case difference between the expected performance of the optimal clairvoyant policy and the expected performance of policy $\pi$:

$$R^{\pi}(T) = \sup_{\mathcal{D}} \left\{ T\mu^* - \mathbb{E}_{\mathcal{D}}^{\pi}\left[\sum_{t=1}^{T} \mu_{\pi_t}\right] \right\}.$$

The *minimax* (optimal) regret of BwSC is defined as $R_S^*(T) = \inf_{\pi \in \Pi_S} R^{\pi}(T)$.

In our paper, when we say a policy is "near-optimal" or "optimal up to logarithmic factors", we mean that its regret bound is optimal in $T$ up to logarithmic factors of $T$, irrespective of whether the bound is optimal in $k$, since typically $k$ is much smaller than $T$ (e.g., $k = O(1)$).

*Remark.* There are two notions of regret in the stochastic bandit literature. The $R^{\pi}(T)$ regret that we consider is called *distribution-free*, as it does not depend on $\mathcal{D}$. On the other hand, one can also define the *distribution-dependent* regret $R_{\mathcal{D}}^{\pi}(T) = T\mu^* - \mathbb{E}_{\mathcal{D}}^{\pi}\left[\sum_{t=1}^{T} \mu_{\pi_t}\right]$ that depends on $\mathcal{D}$. This second notion of regret is only meaningful when $\mu_1, \ldots, \mu_k$ are well-separated. Unlike the classical MAB problem where there are policies simultaneously achieving near-optimal bounds under both regret notions, in the BwSC problem, due to the limited switching budget, finding a policy that simultaneously achieves near-optimal bounds under both regret notions is usually impossible. In this paper, we focus on the distribution-free regret. Extensions to the distribution-dependent regret can be found in the full version of this paper [14].

**Relationship between BwSC and MAB.** Obviously, BwSC and MAB share the same definition of $R^{\pi}(S)$, and the only difference between BwSC and MAB is the existence of a switching constraint $\pi \in \Pi_S$, determined by $(c_{i,j}) \in \overline{\mathbb{R}}_{\geq 0}^{k \times k}$ and $S \in \overline{\mathbb{R}}_{\geq 0}$ (when $S = \infty$, BwSC degenerates to MAB). This makes BwSC a natural framework to study the trade-off between regret and incurred switching cost in MAB. That is, the trade-off between the optimal regret $R_S^*(T)$ and switching budget $S$ in BwSC completely characterizes the trade-off between a policy's best achievable regret and its worst possible incurred switching cost in MAB. We are interested in how $R_S^*(T)$ behaves over a range of switching budget $S$, and how it is affected by the structure of switching costs $(c_{i,j})$.

# 3 Other Related Work

This paper is not the first one to study online learning problems with limited switches. Indeed, a few authors have realized the practical significance of limited switching budget. [10] considers a dynamic pricing model where the demand function is unknown but belongs to a known finite set, and a pricing policy is allowed to make at most $m$ price changes. [9] studies a multi-period stochastic inventory replenishment and pricing problem with unknown parametric demand and limited price changes. We note that both [10, 9] only focus on specific decision-making problems, and their results rely on some strong assumptions about the unknown environment. By contrast, the BwSC model in our paper is generic and assumes no prior knowledge of the environment. The learning task in BwSC is thus more challenging than previous models. In the adversarial setting, [3] studies the adversarial MAB with limited number of switches. Since our problem is stochastic while their problem is adversarial, the results and methodologies in our paper are fundamentally different from their paper. It is worth noting that the switching constraint in BwSC is also more general than the number-of-switch constraints in the above-mentioned models.

The BwSC problem is also related to the batched bandit problem proposed by [13]. The $M$-batched bandit problem is defined as follows: given a classical bandit problem, assumes that the learner must split her learning process into $M$ batches and is only able to observe the realized rewards from a given batch after the entire batch is completed. [13] studies the problem in the case of two arms. Very recently, [11] extends the results to $k$ arms. The batched bandit problem and the BwSC problem are two different problems: the batched bandit problem limits observations and allows unlimited switching, while the BwSC problem limits switching and allows unlimited observations. Surprisingly, we discover some non-trivial connections between the batched bandit problem and the unit-switching-cost BwSC problem, which are presented in the full version of this paper [14].

# 4 Unit Switching Costs

In this section, we consider the BwSC problem with unit switching costs, where $c_{i,j} = 1$ for all $i \neq j$. In this case, since every switch incurs a unit cost, the switching budget $S$ can be interpreted as the maximum number of switches that the learner can make in total. Thus, the unit-switching-cost BwSC problem can be simply interpreted as "MAB with limited number of switches".

## 4.1 Upper Bound on Regret

We first propose a simple and intuitive policy that provides an upper bound on the regret. Our policy, called the $S$-*Switch Successive Elimination* (SS-SE) policy, is described in Algorithm 1. The design philosophy behind the SS-SE policy is to divide the entire horizon into several pre-determined intervals (i.e. batches) and to control the number of switches in each interval. The policy thus has some similarities with the 2-armed batched policy of [13] and the $k$-armed batched policy of [11], which proves to be near-optimal in the batched bandit problem. However, since we are studying a different problem, directly applying a batched policy to the BwSC problem does not work. In particular, in the batched bandit problem, the number of intervals (i.e., batches) is a given constraint, while in the BwSC problem, the switching budget is the given constraint. We thus add two key ingredients into the SS-SE policy: (1) an index $m(S)$ suggesting how many intervals should be used to partition the entire horizon; (2) a switching rule ensuring that the total number of switches within $k$ actions cannot exceed the switching budget $S$. These two ingredients make the SS-SE policy substantially different from an ordinary batched policy.

**Intuition about the policy.** The policy divides the $T$ rounds into $\lfloor \frac{S-1}{k-1} \rfloor + 1$ intervals in advance. The sizes of the intervals are designed to balance the exploration-exploitation trade-off. An active set of "good" actions $A_l$ is maintained for each interval $l$ and at the end of each interval some "bad" actions are eliminated before the start of the next interval. The policy controls the number of switches by ensuring that only $|A_l| - 1$ switches happen within each interval $l$ and at most one switch happens between two consecutive intervals. Finally, in the last interval only the empirical best action is chosen.

We show that the SS-SE policy is indeed an $S$-switching-budget policy and establish the following upper bound on its regret.

**Algorithm 1** $S$-Switch Successive Elimination (SS-SE)

---

**Input:** Number of arms $k$, Switching budget $S$, Horizon $T$

**Partition:** Calculate $m(S) = \left\lfloor \frac{S-1}{k-1} \right\rfloor$.

  Divide the entire time horizon $1, \ldots, T$ into $m(S) + 1$ intervals: $(t_0 : t_1], (t_1 : t_2], \ldots, (t_{m(S)} : t_{m(S)+1}]$, where the endpoints are defined by $t_0 = 0$ and

$$t_i = \left\lfloor k^{1 - \frac{2 - 2^{-(i-1)}}{2 - 2^{-m(S)}}} T^{\frac{2 - 2^{-(i-1)}}{2 - 2^{-m(S)}}} \right\rfloor, \quad \forall i = 1, \ldots, m(S) + 1.$$

**Initialization:** Let the set of all active actions in the $l$-th interval be $A_l$. Set $A_1 = [k]$. Let $a_0$ be a random action in $[k]$.

**Policy:**

1: **for** $l = 1, \ldots, m(S)$ **do**
2:    **if** $a_{t_{l-1}} \in A_l$ **then**
3:        Let $a_{t_{l-1}+1} = a_{t_{l-1}}$. Starting from this action, choose each action in $A_l$ for $\frac{t_l - t_{l-1}}{|A_l|}$ consecutive rounds. Mark the last chosen action as $a_{t_l}$.[1]
4:    **else if** $a_{t_{l-1}} \notin A_l$ **then**
5:        Starting from an arbitrary active action in $A_l$, choose each action in $A_l$ for $\frac{t_l - t_{l-1}}{|A_l|}$ consecutive rounds. Mark the last chosen action as $a_{t_l}$.
6:    **end if**
7:    Statistical test: deactivate all actions $i$ s.t. $\exists$ action $j$ with $\text{UCB}_{t_l}(i) < \text{LCB}_{t_l}(j)$, where

$$\text{UCB}_{t_l}(i) = \text{empirical mean of action } i \text{ in}[1 : t_l] + \sqrt{\frac{2\log T}{\text{number of plays of action } i \text{ in}[1 : t_l]}},$$

$$\text{LCB}_{t_l}(i) = \text{empirical mean of action } i \text{ in}[1 : t_l] - \sqrt{\frac{2\log T}{\text{number of plays of action } i \text{ in}[1 : t_l]}}.$$

8: **end for**
9: In the last interval, choose the action with the highest empirical mean (up to round $t_{m(S)}$).

---

**Theorem 1** *Let $\pi$ be the SS-SE policy, then $\pi \in \Pi_S$. There exists an absolute constant $C \geq 0$ such that for all $k \geq 1$, $S \geq 1$ and $T \geq k$,*

$$R^\pi(T) \leq C(\log k \log T) k^{1 - \frac{1}{2 - 2^{-m(S)}}} T^{\frac{1}{2 - 2^{-m(S)}}},$$

*where $m(S) = \left\lfloor \frac{S-1}{k-1} \right\rfloor$.*

Theorem 1 provides an upper bound on the optimal regret of the unit-switching-cost BwSC problem:

$$R_S^*(T) = \tilde{O}(T^{1/(2 - 2^{-\lfloor (S-1)/(k-1) \rfloor})}).$$

### 4.2 Lower Bound on Regret

The SS-SE policy, though achieves sublinear regret, seems to have many limitations that could have weaken its performance, and on the surface it may suggest that the regret bound is not optimal. We discuss two points here. (1) The SS-SE policy does not make full use of its switching budget. Consider the case of 11 actions and 20 switching budget. Since $m(20) = \lfloor (20 - 1)/(11 - 1) \rfloor = 1 = m(11)$, the SS-SE policy will just run as if it could only make 11 switches, despite the fact that it has 9 additional switching budget (which will never be used). It seems that by tracking and allocating the switching budget in a more careful way, one can achieve lower regret. (2) The SS-SE policy learns from data infrequently. Note that the SS-SE policy pre-determines the number, sizes and locations of its intervals before seeing any data, and executes actions within each interval based on a pre-determined schedule. Consider again the case of 11 actions and 20 switching budget, the SS-SE policy will split the entire horizon into two intervals and will only learn from data at the end of the

first interval, after which it will choose a single action to be applied throughout the entire second interval. It seems that by learning from data more frequently, one can achieve lower regret.

While the above arguments are based on our first instinct and seem very reasonable, surprisingly, all of them prove to be wrong: no $S$-switch policy can theoretically do better! In fact, we match the upper bound provided by SS-SE by showing an information-theoretic lower bound in Theorem 2. This indicates that the SS-SE policy is rate-optimal up to logarithmic factors, and $R_S^*(T) = \tilde{\Theta}(T^{1/(2-2^{-\lfloor (S-1)/(k-1)\rfloor})})$. Note that the tightness of $T$ is acheived *per instance*, i.e., for every $k$ and every $S$. That is, our lower bound is substantially stronger than a single lower bound demonstrated for specific $k$ and $S$. The proof of the lower bound involves a novel "tracking the cover time" argument that (to the best of our knowledge) has not appeared in previous literature and may be of independent interest. We state the lower bound and give a sketch of the proof below.

**Theorem 2** *There exists an absolute constant $C > 0$ such that for all $k \geq 1, S \geq 1, T \geq k$ and for all policy $\pi \in \Pi_S$,*

$$R^\pi(T) \geq \begin{cases} C \left( k^{-\frac{3}{2} - \frac{1}{2-2^{-m(S)}}} (m(S)+1)^{-2} \right) T^{\frac{1}{2-2^{-m(S)}}}, & \text{if } m(S) \leq \log_2 \log_2(T/k), \\ C\sqrt{kT}, & \text{if } m(S) > \log_2 \log_2(T/k), \end{cases}$$

*where $m(S) = \left\lfloor \frac{S-1}{k-1} \right\rfloor$.*

**Proof idea.** For any $k \geq 1, S \geq 1$ and $T \geq k$, for any $S$-switch policy $\pi \in \Pi_S$, we want to find an environment $\mathcal{D}$ such that $R_{\mathcal{D}}^\pi(T)$ is larger than the desired lower bound. A key challenge here is that $\pi$ is an arbitrary and abstract $S$-switch policy — we need more information about $\pi$ to construct $\mathcal{D}$. With this goal in mind, we first design a concrete "primal environment" $\alpha$. We use this environment to evaluate policy $\pi$, such that we can observe some key patterns revealed by policy $\pi$ under $\alpha$. These patterns are characterized by a series of ordered stopping times $\tau_1 \leq \tau_2 \leq \cdots \leq \tau_{m(S)+1}$, some of which may be $\infty$, that are recursively defined as follows:

- $\tau_1$ is the first time that all the actions in $[k]$ have been chosen in period $[1 : \tau_1]$,
- $\tau_2$ is the first time that all the actions in $[k]$ have been chosen in period $[\tau_1 : \tau_2]$,
- Generally, $\tau_i$ is the first time that all the actions in $[k]$ have been chosen in period $[\tau_{i-1} : \tau_i]$, for $i = 2, \ldots, m(S)+1$.

We then compare the realization of $\tau_1, \ldots, \tau_{m(S)}$ with a series of fixed values $t_1, \ldots, t_{m(S)}$, which are the endpoints of the intervals defined in Algorithm 1. Based on the possible outcomes of comparisons, we define $m(S)+1$ key events:

- $E_1 = \{\tau_1 > t_1\}$,
- $E_j = \{\tau_{j-1} \leq t_{j-1}, \tau_j > t_j\}$, for $j = 2, \ldots, m(S)$,
- $E_{m(S)+1} = \{\tau_{m(S)} \leq t_{m(S)}\}$,

at least one of which must occur under $\pi$ and $\alpha$ with probability at least $1/(m(S)+1)$. We then do a case by case analysis as follows. In the first case, $\{\tau_1 > t_1\}$ occurs with certain probability, indicating that the action chosen in round $\tau_1$ was not chosen in $[1 : t_1]$ with certain probability; in the second case, $\exists j \in [2 : m(S)]$ such that $\{\tau_{j-1} \leq t_{j-1}, \tau_j > t_j\}$ occurs with certain probability, indicating that the action chosen in round $\tau_j$ was not chosen in $[t_{j-1} : t_j]$ with certain probability; in the third case, $\{\tau_{m(S)} \leq t_{m(S)}\}$ with certain probability, indicating that the number of switches occurs in $[t_{m(S)} : T]$ is at most $k - 1$. For each case, we construct an "auxiliary environment" $\beta$ by carefully adjusting $\alpha$ based on the aforementioned indication. The environment $\beta$ ensures two things: (1) $\beta$ is "hard for $\pi$ to distinguish from $\alpha$", such that a crucial event $\mathcal{E}$ (constructed based on the indication) that occurs under $\pi$ and $\alpha$ with certain probability also occurs under $\pi$ and $\beta$ with similar probability; and (2) $\beta$ is "different enough from $\alpha$" such that the certain occurrence probability of the event $\mathcal{E}$ under $\beta$ makes $R_\beta^\pi(T)$ larger than the desired lower bound. Theorem 2 then follows by $R^\pi(T) \geq R_\beta^\pi(T)$. For the complete proof of Theorem 2, see the full version of this paper [14].

Combining Theorem 1 and Theorem 2, we have

**Corollary 1** *For any fixed $k \geq 1$, for any $S \geq 1$, $R_S^*(T) = \tilde{\Theta}(T^{1/(2-2^{-\lfloor (S-1)/(k-1)\rfloor})})$.*

*Remark.* We briefly explain why the upper and lower bounds in Theorem 1 and Theorem 2 match in $T$. When $m(S) \leq \log_2 \log_2(T/k)$, which is the case we are mostly interested in, $(m(S)+1)^2 = o(\log T)$, thus the upper and lower bounds match within $o((\log T)^2)$. When $m(S) > \log_2 \log_2(T/k)$, the upper bound is $O(\sqrt{T} \log T)$, thus the upper and lower bounds directly match within $O(\log T)$.

## 4.3 Phase Transitions and Cyclic Phenomena

Corollary 1 allows us to characterize the trade-off between the switching budget $S$ and the optimal regret $R_S^*(T)$. To illustrate this trade-off, Table 1 depicts the behavior of $R_S^*(T)$ as a function of $S$ given a fixed $k$. Note that as discussed in Section 2, the relationship between $R_S^*(T)$ and $S$ also characterizes the inherent trade-off between regret and maximum number of switches in the classical MAB problem.

Table 1: Regret as a Function of Switching Budget

| $S$ | $[0, k)$ | $[k, 2k-1)$ | $[2k-1, 3k-2)$ | $[3k-2, 4k-3)$ | $[4k-3, 5k-4)$ |
|---|---|---|---|---|---|
| $R_S^*(T)$ | $\tilde{\Theta}(T)$ | $\tilde{\Theta}(T^{2/3})$ | $\tilde{\Theta}(T^{4/7})$ | $\tilde{\Theta}(T^{8/15})$ | $\tilde{\Theta}(T^{16/31})$ |
| $R_S^*(T)/R_\infty^*(T)$ | $\tilde{\Theta}(T^{1/2})$ | $\tilde{\Theta}(T^{1/6})$ | $\tilde{\Theta}(T^{1/14})$ | $\tilde{\Theta}(T^{1/30})$ | $\tilde{\Theta}(T^{1/62})$ |

As we have shown, $R_S^*(T) = \tilde{\Theta}(T^{1/(2-2^{-\lfloor (S-1)/(k-1) \rfloor})})$. To the best of knowledge, this is the first time that a floor function naturally arises in the order of $T$ in the optimal regret of an online learning problem. As a direct consequence of this floor function, we discover several surprising phenomena regarding the trade-off between $S$ and $R_S^*(T)$ for any given $k$.

**Definition 2** *(Phases and Transition Points) For a $k$-armed unit-switching-cost BwSC, we call the interval $[(j-1)(k-1)+1, j(k-1)+1)$ the j-th* phase*, and call $j(k-1)+1$ the j-th* transition point *($j \in \mathbb{Z}_{>0}$).*

**Fact 1** *(Phase Transitions) As $S$ increases from $0$ to $\Theta(\log \log T)$, $S$ will leave the j-th phase and enter the $(j+1)$-th phase at the j-th transition point ($j \in \mathbb{Z}_{>0}$). Each time $S$ arrives at a transition point, $R_S^*(T)$ will drop significantly, and stay at the same level until $S$ arrives the next transition point.*

**Fact 2** *(Cyclic Phenomena) The length of each phase is always equal to $k-1$, independent of $S$ and $T$. We call the quantity $k-1$ the* budget cycle*, which is the length of each phase.*

Phase transitions are clearly presented in Table 1. This phenomenon seems counter-intuitive, as it suggests that increasing switching budget would not help to decrease the best achievable regret, as long as the budget does not reach the next transition point. Note that phase transitions are only exhibited when $S$ is in the range of $0$ to $\Theta(\log \log T)$. After $S$ exceeds $\Theta(\log \log T)$, $R_S^*(T)$ will reamin unchanged at the level of $\tilde{\Theta}(\sqrt{T})$ — the optimal regret will only vary within logarithmic factors and there is no significant regret drop any more. Therefore, one can also view $\Theta(\log \log T)$ as a "final transition point" that marks the disappearance of phase transitions.

Cyclic Phenomena indicate that, assuming that the learner's switching budget is at a transition point, then the extra switching budget that the learner needs to achieve the next regret drop (i.e., to arrive at the next transition point) is always $k-1$. Cyclic phenomena also seem counter-intuitive: when the learner has more switching budget, she can conduct more statistical tests, eliminate more bad actions (which can be thought of as reducing $k$) and allocate her switching budget in a more flexible way — all of these suggest that the budget cycle should be a quantity decreasing with $S$. However, the cyclic phenomena tell us that the budget cycle is always a constant and no learning policy in the unit-cost BwSC (and in MAB) can escape this cycle, no matter how large $S$ is, as long as $S = o(\log \log T)$.

On the other hand, as $S$ contains more and more budget cycles, the gap between $R_S^*(T)$ and $R_\infty^*(T) = \tilde{\Theta}(\sqrt{T})$ does decrease dramatically. In fact, $R_S^*(T)$ decreases *doubly exponentially* fast as $S$ contains more budget cycles. From Table 1, we can verify that 3 or 4 budget cycles are already enough for an $S$-switching-budget policy to achieve close-to-optimal regret in MAB (compared with the optimal policy with unlimited switching budget).

Finally, we give some comments on the scope of our results. Note that phase transitions and cyclic phenomena are associated with theoretical bounds of the worst-case regret, so if (1) the underlying distributions are not the worst-case distributions and we are focusing on the "actual incurred regret", or (2) $T$ is not large enough to dominate the constants in the bounds, phase transitions and cyclic phenemona may not be exhibited.

## 5 General Switching Costs

We now proceed to the general case of BwSC, where $c_{i,j}$ $(= c_{j,i})$ can be any non-negative real number and even $\infty$. The problem is significantly more challenging in this general setting. For this purpose, we need to enhance the framework of Section 2 to better characterize the structure of switching costs. We do this by representing switching costs via a weighted graph.

Let $G = (V, E)$ be a (weighted) complete graph, where $V = [k]$ (i.e., each vertex corresponds to an action), and the edge between $i$ and $j$ is assigned a weight $c_{i,j}$ $(\forall i \neq j)$. We call the weighted graph $G$ the *switching graph*. In this paper, we assume the switching costs satisfy the triangle inequality: $\forall i, j, l \in [k], c_{i,j} \leq c_{i,l} + c_{l,j}$.

The results of the unit-switching-cost model suggest that an effective policy that minimizes the worst-case regret must repeatedly visit all actions, in a manner similar to the SS-SE policy. This indicates that in the general-switching-cost model, an effective policy should repeatedly visit all vertices in the switching graph, in a most economical way to stay within budget. Motivated by this idea, we propose the *Hamiltonian-Switching Successive Elimination* (HS-SE) policy, and present its details in Algorithm 2. The HS-SE policy enhances the original SS-SE policy by adding two additional ingredients: (1) a pre-specified switching order: within each interval, the HS-SE policy switches based on an order determined by the shortest Hamiltonian path of the switching graph $G$; (2) a reversing policy: the HS-SE policy switches along one direction in the odd intervals, and along the reverse direction in the even intervals. Note that while the shortest Hamiltonian path problem is NP-hard, solving this problem is entirely an "offline" step in the HS-SE policy. That is, for a given switching graph, the learner only needs to solve this problem once.

Let $H$ denote the total weight of the shortest Hamiltonian path of $G$. We give an upper bound on the regret of the HS-SE policy in Theorem 3.

**Theorem 3** *Let $\pi$ be the HS-SE policy, then $\pi \in \Pi_S$. There exists an absolute constant $C \geq 0$ such that for all $G$, $k = |G|$, $S \geq 0$, $T \geq k$,*

$$R^\pi(T) \leq C(\log k \log T)k^{1 - \frac{1}{2 - 2^{-m_G^U(S)}}} T^{\frac{1}{2 - 2^{-m_G^U(S)}}},$$

*where $m_G^U(S) = \left\lfloor \frac{S - \max_{i,j \in [k]} c_{i,j}}{H} \right\rfloor$.*

We then give a lower bound that is close to the above upper bound, see Theorem 4. Compared to the proof of Theorem 2 (see Section 4.2 for a proof sketch), we would like to highlight an important new step in the proof of Theorem 4. Recall that in the proof sketch of Theorem 2, we mention a step of constructing the "primal environment" $\alpha$. In the proof of Theorem 4, our construction of $\alpha$ ensures that $\alpha$ has an additional property: $(\arg \max_{i \in [k]} \min_{j \neq i} c_{i,j})$ is the optimal action in $\alpha$. This property makes $(\max_{i \in [k]} \min_{j \neq i} c_{i,j})$ a lower bound on the cost incurred by switching between a sub-optimal action and the optimal action in $\alpha$. Our new proof utilizes this property and makes the quantity $(\max_{i \in [k]} \min_{j \neq i} c_{i,j})$ appear in the lower bound. For the complete proof of Theorem 4, see the full version of this paper [14].

**Theorem 4** *There exists an absolute constant $C > 0$ such that for all $G, k = |G|, S \geq 0, T \geq k$ and for all policy $\pi \in \Pi_S$,*

$$R^\pi(T) \geq \begin{cases} C\left(k^{-\frac{3}{2} - \frac{1}{2 - 2^{-m_G^L(S)}}} (m_G(S) + 1)^{-2}\right) T^{\frac{1}{2 - 2^{-m_G^L(S)}}}, & \text{if } m_G^L(S) \leq \log_2 \log_2(T/k), \\ C\sqrt{kT}, & \text{if } m_G^L(S) > \log_2 \log_2(T/k), \end{cases}$$

*where $m_G^L(S) = \left\lfloor \frac{S - \max_{i \in [k]} \min_{j \neq i} c_{i,j}}{H} \right\rfloor$.*

---

**Algorithm 2** Hamiltonian-Switching Successive Elimination (HS-SE)

---

**Input:** Switching Graph $G$, Switching budget $S$, Horizon $T$

**Offline Step:** Find the shortest Hamiltonian path in $G$: $i_1 \to \cdots \to i_k$. Denote the total weight of the shortest Hamiltonian path as $H$. Calculate $m_G^U(S) = \left\lfloor \frac{S - \max_{i,j \in [k]} c_{i,j}}{H} \right\rfloor$.

**Partition:** Run the partition step in the SS-SE policy with $m(S) = m_G^U(S)$.

**Initialization:** Let the set of all active actions in the $l$-th interval be $A_l$. Set $A_1 = [k]$, $a_0 = i_1$.

**Policy:**

1: **for** $l = 1, \ldots, m_G^U(S)$ **do**
2:     **if** $a_{t_{l-1}} \in A_l$ and $l$ is odd **then**
3:         Let $a_{t_{l-1}+1} = a_{t_{l-1}}$. Starting from this action, along the direction of $i_1 \to \cdots \to i_k$, choose each action in $A_l$ for $\frac{t_l - t_{l-1}}{|A_l|}$ consecutive rounds. Mark the last chosen action as $a_{t_l}$.
4:     **else if** $a_{t_{l-1}} \in A_l$ and $l$ is even **then**
5:         Let $a_{t_{l-1}+1} = a_{t_{l-1}}$. Starting from this action, along the direction of $i_k \to \cdots \to i_1$, choose each action in $A_l$ for $\frac{t_l - t_{l-1}}{|A_l|}$ consecutive rounds. Mark the last chosen action as $a_{t_l}$.
6:     **else if** $a_{t_{l-1}} \notin A_l$ and $l$ is odd **then**
7:         Along the direction of $i_1 \to \cdots \to i_k$, find the first action that still remains in $A_l$. Starting from this action, along the direction of $i_1 \to \cdots \to i_k$, choose each action in $A_l$ for $\frac{t_l - t_{l-1}}{|A_l|}$ consecutive rounds. Mark the last chosen action as $a_{t_l}$.
8:     **else if** $a_{t_{l-1}} \notin A_l$ and $l$ is even **then**
9:         Along the direction of $i_k \to \cdots \to i_1$, find the first action that still remains in $A_l$. Starting from this action, along the direction of $i_k \to \cdots \to i_1$, choose each action in $A_l$ for $\frac{t_l - t_{l-1}}{|A_l|}$ consecutive rounds. Mark the last chosen action as $a_{t_l}$.
10:    **end if**
11:    Statistical test: deactivate all actions $i$ s.t. $\exists$ action $j$ with $\text{UCB}_{t_l}(i) < \text{LCB}_{t_l}(j)$, where

$$\text{UCB}_{t_l}(i) = \text{empirical mean of action } i \text{ in}[1:t_l] + \sqrt{\frac{2 \log T}{\text{number of plays of action } i \text{ in}[1:t_l]}},$$

$$\text{LCB}_{t_l}(i) = \text{empirical mean of action } i \text{ in}[1:t_l] - \sqrt{\frac{2 \log T}{\text{number of plays of action } i \text{ in}[1:t_l]}}.$$

12: **end for**
13: In the last interval, choose the action with the highest empirical mean (up to round $t_{m_G^U(S)}$).

---

Finally, we illustrate how tight the above upper and lower bounds are. When the switching costs satisfy the condition $\max_{i,j \in [k]} c_{i,j} = \max_{i \in [k]} \min_{j \neq i} c_{i,j}$, the two bounds directly match. When this condition is not satisfied, for any switching graph $G$, the above two bounds still match for a wide range of $S$:

$$\left[ 0, H + \max_{i \in [k]} \min_{j \neq i} c_{i,j} \right) \bigcup \left\{ \bigcup_{n=1}^{\infty} \left[ nH + \max_{i,j \in [k]} c_{i,j}, (n+1)H + \max_{i \in [k]} \min_{j \neq i} c_{i,j} \right) \right\}.$$

Even when $S$ is not in this range, we still have $m_G^U(S) \leq m_G^L(S) \leq m_G^U(S) + 1$ for any $G$ and any $S$, which means that the difference between the two indices is at most 1 and the regret bounds are always very close. In fact, it can be shown that as $S$ increases, the gap between the upper and lower bounds decreases *doubly exponentially*. Therefore, the HS-SE policy is quite effective for the general BwSC problem.

For additional theoretical results for the general BwSC problem, see the full version of this paper [14].

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
