[Reviews · NeurIPS 2019]

Reviewer 1



Relation to the Batched bandit problem: the authors claim that the relationship between the switching cost constrained MAB problem and the batched bandit problem as described in lines 143-154 is somewhat surprising. While the two problems are indeed different, this relation is not really that surprising. In fact there is a well known relationship between mini-batched bandit algorithms and bandit algorithms with switching costs in the adversarial setting. In particular, showing a lower bound for the MAB with switching costs for a certain type of mini-batched algorithms goes through a lower bound for the adversarial MAB with switching costs (see for example Bandits with Switching Costs: T^{2/3} Regret). The main idea behind the algorithm is to carefully divide the T rounds of the MAB game into segments. Before each segment a set of active arms is pruned according to a UCB/LCB strategy and then the arms from the active set are played an equal amount of times in the segment. The crux of the proof is choosing the length of each segment according to the bound on number of switches and number of arms. While the batch sizes induced by each segment might be novel, the proof seems standard. The main technical contributions begin with the lower bound proof of Theorem 2. The proof seems non-trivial and the use of stopping times according to when each arm has been pulled for the first time to my knowledge is novel. Since the proof of the lower bound is an important contribution of this paper, the paper might benefit from a lengthier explanation about how the proof works exactly or at least some more intuition as the paragraph in between lines 194-204 is not really helpful when reading the Appendix. For example the authors might want to explain more carefully how the stopping times come into the proof, as a case by case analysis and how the auxiliary policy \beta is constructed to show the lower bound. While the discussion about the Phase transitions and Cyclic Phenomena is insightful, the space in the main paper might be better used to expand on the proof of Theorem 2. The main paper will also benefit from a lengthier discussion about Algorithm 3 which is deferred to the appendix and the proofs of Theorems 3 and 4 as the proofs of these theorems seem non-trivial and a good contribution. Can the authors also clarify, how the bounds in Theorem 3 will change if the shortest Hamiltonian path is only approximately computed in Algorithm 3, instead of exactly? Overall the contributions of the paper seem significant and especially the proof of the lower bounds seem novel. The relation between arbitrary switching costs and the shortest Hamiltonian path is also maybe surprising. The paper’s presentation can be improved if the authors spend less time discussing implications of Theorem 1 and Theorem 2 and focus more on the proof techniques and Algorithm 3, which are currently deferred to the appendix. ----------------------------------------------------------------------------------------- I have read the response by the authors and other reviews. I am mostly satisfied with the rebuttal and the proposed reorganization of the paper. While I agree with the comments of other reviewers and it would definitely be nice to have a more thorough treatment of the phase transitions, I find that the rebuttal addresses most of the raised concerns and will keep my current score.

Reviewer 2



Originality: The scope of the paper is higher original in both achieving and discussing a regret bound with multiple phase changes. Notably this is achieved in a setting well motivated in literature and with a model corresponding to real use cases. Clarity: The paper is well written. The language is easy to read and informative while still being sufficiently explicit for instance when defining algorithms. While what is written is excellent the space management of the paper is decidedly poor. In my opinion this is the paper's main weakness: While the abstract and introduction promises treatment of the case of general switching costs and connections to graph traversal, this significant part of the paper is given half a page (section 5) with large portions of not just technicalities relegated to appendices. From my point of view this has not received a proper treatment in the main text of the paper, and I will largely consider it outside of the scope of the present text! With this removed it is however not unrealistic that the remaining scope could be accepted! Similarly it is shame that little space is devoted to proof techniques in the main text. Significance: The results are important in two ways: The model of external switching constraints is a significant framework for dealing with real life models. Secondly achieving a regret with phase transitions in this fashion is interesting in its own right and the paper puts a great deal of effort into the discussion hereof, which could spark interesting discussions of characterisations of bandits in general. -- Further comments and questions: 1) It would be nice with a more thorough treatment of the necessity of the phase transitions in the "actual regret scaling" compared to the asymptotic bounds shown here. In other words answering the question: Could there be a smooth regret scaling (i.e. not displaying phase transitions) for which the proven bounds still holds? 2) While the even spacing of phases is interesting, the terminology of "cyclical" seems a bit contrived. It is technically correct but seems oversold as a separat topic in the title rather than a characterisation of the phase transitions. ==== After the author response and reading the other reviews, my opinion is that the paper has significant contributions and the authors are willing and able to significantly reorganise the paper in a way to address our concerns. While a thorough reorganisation plan is sketched out in the response, it is my opinion that this is too significant of a change to accept without further review.

Reviewer 3



This is a novel result on the MAB problem with limited switch cost problem. The authors show a intuitive method to reduce the switch cost in the unit swtich cost case, and then prove that their method is optimal. They also show the constraint on the number of switches inside an MAB game. The regret is reduced a lot only when the budget of the switches has been larger by k-1, which is the minimum cost to try every arm in a period. Then they generalize their result to the case that the switch costs are arbitrary. In this case, and propose the HS-SE policy. This policy is still optimal in a lot of kinds of swtich cost case, and near optimal in other cases. This is a good paper that shows the relationship between number of switches and the culmulative regret. It is useful in some kind of real applications. The proofs are correct, and the writting is clear as well.

[Author Response · NeurIPS 2019]

Thanks for your insightful comments and for your recognition of the originality (reviewers 1, 2 & 3) and significance (reviewers 1 & 2) of our paper. Below we respond to your comments.

**To Reviewer 1 & 2: Reorganization Plan.** We agree that the paper's space management should be improved. Based on your comments, we propose the following feasible reorganization plan: (1) provide more explanations on the lower bound proof techniques; (2) increase Section 5 (general switching costs) to about 1.5 pages with more detailed discussions of the HS-SE policy and Theorem 3 & 4; (3) revise the abstract & introduction to highlight the contributions that would be discussed in detail in the main text; and (4) remove some secondary content.

**(1) Expanded proof sketch of Theorem 2.** For any $T \geq 1, k \geq 1$ and $S \geq 0$, for any $S$-switch policy $\pi \in \Pi_S$, we want to find an environment $\mathcal{D}$ such that $R_{\mathcal{D}}^\pi(T)$ is larger than the desired lower bound. A key challenge here is that $\pi$ is an arbitrary and abstract $S$-switch policy — we need more information about $\pi$ to construct $\mathcal{D}$. With this goal in mind, we first design a concrete "primal environment" $\alpha$. We use this environment to evaluate policy $\pi$, such that we can observe some key patterns revealed by policy $\pi$ under $\alpha$. These patterns are characterized by a series of ordered stopping times $\tau_1 \leq \tau_2 \leq \ldots \leq \tau_{m(S)+1}$, some of which may be $\infty$, that are recursively defined as follows: add line 197-200 of the article. We then compare the realization of $\tau_1, \ldots, \tau_{m(S)}$ with a series of fixed values $t_1, \ldots, t_{m(S)}$, which are the endpoints of the intervals defined in Algorithm 1. Based on the possible outcomes of comparisons, we define $m(S) + 1$ key events: add line 9-11 in Appendix page 5, at least one of which must occur under $\pi$ and $\alpha$ with probability at least $1/(m(S) + 1)$. We then do a case by case analysis as follows. In the first case, $\{\tau_1 > t_1\}$ occurs with certain probability, indicating that the action chosen in round $\tau_1$ was not chosen in $[1 : t_1]$ with certain probability; in the second case, $\exists j \in [2 : m(S)]$ such that $\{\tau_{j-1} \leq t_{j-1}, \tau_j > t_j\}$ occurs with certain probability, indicating that the action chosen in round $\tau_j$ was not chosen in $[t_{j-1} : t_j]$ with certain probability; in the third case, $\{\tau_{m(S)} \leq t_{m(S)}\}$ with certain probability, indicating that the number of switches occurs in $[t_{m(S)} : T]$ is at most $k - 1$. For each case, we construct an "auxiliary environment" $\beta$ by carefully adjusting $\alpha$ based on the aforementioned indication. The environment $\beta$ ensures two things: (i) $\beta$ is "hard for $\pi$ to distinguish from $\alpha$", such that a crucial event (constructed based on the indication) that occurs under $\pi$ and $\alpha$ with certain probability also occurs under $\pi$ and $\beta$ with similar probability; and (ii) $\beta$ is "different enough from $\alpha$" such that the certain occurrence probability of this event under $\beta$ makes $R_\beta^\pi(T)$ larger than the desired lower bound. Theorem 2 then follows by $R^\pi(T) \geq R_\beta^\pi(T)$.

**(2) Increasing Section 5.** First, we will have a lengthier discussion of the HS-SE policy. We emphasize and explain two new ingredients of the HS-SE policy that are not in the SS-SE policy. (i) A pre-specified switching order: within each interval, the HS-SE policy switches based on an order determined by the shortest Hamiltonian path of the switching graph $G$; (ii) A reversing policy: the HS-SE policy switches along one direction in the odd intervals, and along the reverse direction in the even intervals. We then illustrate how (i) and (ii) enable the HS-SE policy to repeatedly visit all effective actions in an economical way to stay within budget, and how this motivates Theorem 3. Second, we will provide insights on how we extend the unit-cost lower bound to the general-cost lower bound, by highlighting an important step of the proof, which is to let $(\arg\max_i \min_{j \neq i} c_{i,j})$ be the optimal action in the "primal environment" $\alpha$ so that switching from and to this action is costly. Third, we briefly discuss the implications of the bounds.

**(3) Revising abstract & introduction.** We will adjust the abstract and introduction to clarify the focus and scope of the paper. In particular, we will highlight the following contributions: (i) the SS-SE policy, the lower bound proof, and phase transitions and cyclic phenomena in the unit-cost setting; and (ii) the HS-SE policy and extended bounds in the general-cost setting, with a surprising connection to the shortest Hamiltonian path problem.

**(4) Removing some content.** We defer Figure 1 (as Table 1 is enough), Section 4.3.2, and Section 4.4 to the appendix, and greatly shorten line 65-85 & 115-131. Then we have enough space to expand the proof sketch and Section 5.

**To Reviewer 1.** (1) We agree that given the related literature in the adversarial MAB, a "conceptual" relationship between BwSC and the batched bandit problem in the stochastic setting may be expected. However, what we discover is a precise relationship — a "regret equivalence" between $S$-switch $k$-armed BwSC and the $M$-batched $k$-armed bandit problem, see line 275-278. That is, there is an explicit formula $M = \lfloor \frac{S-1}{k-1} \rfloor$ directly translating the regret bounds and the optimal algorithms of the two problems. While this is a surprise to us, we follow your suggestion and defer Section 4.4 to the appendix. (2) If the shortest Hamiltonian path is only approximately computed in Algorithm 3, say we have a Hamiltonian path of length $\delta H$ ($\delta > 1$) instead of the shortest Hamiltonian path of length $H$, then the index $m_G^U(S)$ in the upper bound in Theorem 3 becomes $\lfloor \frac{S - \max_{i,j} c_{i,j}}{\delta H} \rfloor$. As long as $S$ is not too small, the new upper bound is still close to the lower bound in Theorem 4 — the gap between them decreases doubly exponentially with $S$.

**To Reviewer 2.** Yes, the "actual regret scaling" could be smoother than the "worst-case regret scaling". Note that phase transitions are associated with asymptotic bounds of the worst-case regret, so if (1) the underlying distributions are not the worst-case distributions and we are focusing on the "actual incurred regret", or (2) $T$ is not large enough to dominate the constants in the bounds, phase transitions may not be exhibited. We will highlight point (1) and (2) in Section 4.3.1.

**To Reviewer 3.** Following your suggestion, we conducted computational experiments in the setting of $k = 3$, $S = 1, \ldots, 6$ and $T = 10^3, 2 \times 10^3, \ldots, 10^4$. Running the SS-SE policy under several sets of underlying distributions, we, as expected, observe the cyclic phenomena of the incurred regret. However, it is computationally expensive to show the cyclic phenomena of the worst-case regret of other policies, as this requires iterating over all possible distributions.

[Meta-Review · NeurIPS 2019]

To be frank, this paper falls squarely on the acceptance boundary. The decision to accept it is based on the convincing author rebuttal, in particular the authors' willingness to improve the paper based on the reviewer suggestions, as witnessed by the restructuring plan presented in the author rebuttal. While it would have been better to re-review the revised paper, there is neither process nor capacity for this at NeurIPS. Yet given the fact that the results are interesting, I think the conference will benefit from allowing the authors to execute their revision proposal.